# Protocol for LAsting Symptoms after Oesophageal Resectional Surgery (LASORS): multicentre validation cohort study

Sheraz Rehan Markar [1], Ewen A Griffiths,[2] Paul Behrens,[3] Pritam Singh,[4] Ravi S Vohra,[4] James Gossage,[5] Tim Underwood,[6] George B Hanna,[7] On behalf of the LASORS study group

For numbered affiliations see end of article.

**Correspondence to**
Dr Sheraz Rehan Markar;
s.markar@imperial.ac.uk

## ABSTRACT

**Introduction** Surgery is the primary curative treatment for oesophageal cancer, with considerable recent improvements in long-term survival. However, surgery has a long-lasting impact on patient's health-related quality of life (HRQOL). Through a multicentre European study, our research group was able to identify key symptoms that affect patient's HRQOL. These symptoms were combined to produce a tool to identify poor HRQOL following oesophagectomy (LAsting Symptoms after Oesophageal Resection (LASOR) tool). The objective of this multicentre study is to validate a six-symptom clinical tool to identify patients with poor HRQOL for use in everyday clinical practice.

**Methods and analysis** Included patients will: (1) be aged 18 years or older, (2) have undergone an oesophagectomy for cancer between 2015 and 2019, and (3) be at least 12 months after the completion of adjuvant oncological treatments. Patients will be given the previously created LASOR questionnaire. Each symptom from the LASOR questionnaire will be graded according to impact on quality of life and frequency of the symptom, with a composite score from 0 to 5. The previously developed LASOR symptom tool will be validated against HRQOL as measured by the European Organisation for Research and Treatment of Cancer QLQC30 and OG25.

**Sample size** With a predicted prevalence of poor HRQOL of 45%, based on the previously generated LASOR clinical symptom tool, to validate this tool with a sensitivity and specificity of 80%, respectively, a minimum of 640 patients will need to be recruited to the study.

**Ethics and dissemination** NHS Health Research Authority (North East—York Research Ethics Committee) approval was gained 8 November 2019 (REC reference 19/NE/0352). Multiple platforms will be used for the dissemination of the research data, including international clinical and patient group presentations and publication of research outputs in a high impact clinical journal.

## INTRODUCTION

Globally, oesophageal cancer is the 12th most common cancer type and the 7th most common cause of cancer-related death, with an overall 5-year survival of less than

**Strengths and limitations of this study**

► Large, multicentre, contemporenous, European study of quality of life after esophagectomy.
► Comparison will be performed with more comprehensive quality of life tools (European Organisation for Research and Treatment of Cancer-QLQC30 and OG25).
► Quality of Life Questionnaire/LAsting Symptoms after Oesophageal Resection (LASOR) tool has been developed in association with patient with oesophageal cancer support group charities (Heartburn Cancer UK and Oesophageal Patients Association).
► The LASOR tool will be validated in patients who have had potentially curative oesophageal surgery within 1–4 years and could suffer some bias of well-motivated patients responding to invites and some patients suffering from symptoms of undiagnosed recurrent cancer being included.
► Patients who have already suffered recurrent cancer or early postoperative death will not be included.

20%.[1–3] The mainstay of curative treatment for oesophageal cancer is surgical resection, which is often combined with chemotherapy or chemoradiotherapy.[4 5] This multimodality approach to treatment, along with centralisation of oesophageal cancer surgery to high-volume centres and the introduction of minimally invasive approaches to surgery, has been associated with major recent improvements in short-term and oncological outcomes and survival.[6–8]

With the recent improvements in survival, the assessment of patient-reported outcome measures including patient's health-related quality of life (HRQOL) and the impact of long-term symptoms in survivorship has become increasingly important. However, current data regarding the long-term symptom burden following potentially curative oesophagectomy are limited

and evidence-based interventions are lacking. One population-based cohort study suggested approximately 40% of patients seek medical attention for long-term symptoms, associated with increased depression and anxiety.[9]

The European Organisation for Research and Treatment of Cancer (EORTC) have developed questionnaires for the assessment of HRQOL during treatment for oesophageal cancer.[10 11] These tools have been widely used in a research setting to assess the impact on HRQOL of multimodality treatment, variations in surgical technique and complications.[12 13] However, the EORTC-QLQC30 and OG25 modules are often considered too cumbersome to be routinely used in clinical practice. Furthermore, despite frequently being used in cancer-free survivorship, these questionnaires are not designed for this purpose, when the nature of symptoms and their HRQOL impact may be significantly different, compared with the context of diagnosis and treatment.

Through a multicentre European study of 876 patients, we recently identified three key symptoms that were independently associated with poor HRQOL as measured by validated EORTC tools.[14] We presented these findings to the Oesophageal Patient Association (UK) and Heartburn Cancer UK patient support groups who identified a further three symptoms to be included in the final clinical symptom tool that we aim to validate in this present UK study.

## OBJECTIVES
### Global objective
► To validate a six-symptom clinical tool to identify patients with poor HRQOL more than 1 year after surgery, for use in everyday clinical practice.

### Specific objectives
1. Validate a six symptom clinical tool to identify patients with poor HRQOL as measured by current EORTC QLQC30 and OG25 tools.

2. To assess patient acceptability of this clinical symptom-based tool for postoperative clinical follow-up.

## METHODS AND ANALYSIS
### Study design
Cross-sectional study to validate a symptom questionnaire in the identification of patients with poor HRQOL following oesophageal cancer surgery. Eligible patients from the UK will be invited to complete the LAsting Symptoms after Oesophageal Resection (LASOR) tool (table 1) and the EORTC QLQC30 and OG25 questionnaires. Centres from the existing Association of Upper Gastro-intestinal Surgeons for Great Britain and Ireland (AUGIS) research network will be contacted for inclusion in the study, and patients will be consented locally to participate in the study (online supplementary appendix A). The study will begin in October 2020 and run till October 2021 for patient recruitment, with analysis completed and ready for presentation and publication in December 2021. Patients will be sent the questionnaire electronically or in paper form based on their preference indicated at the time of consent to complete the questionnaire at home and avoid reporting bias. Patients who fail to respond or complete the questionnaire will receive one telephone or electronic reminder and if they fail to respond will be excluded from the study.

### Measures
#### Archival data
This data will be collected retrospectively from the patient medical records (online supplementary appendix B).

#### Demographic information
Data on pretreatment; patient age, sex, body mass index, ethnic background, socioeconomic status (Carstairs index), education level, smoking status and medical comorbidities (collated with Charlson Comorbidity Index) will be collected through review of medical records.

| Table 1 | LAsting Symptoms after Oesophageal Resection symptom tool | | | | | | | |
| --- | --- | --- | --- | --- | --- | --- | --- | --- |
| | **Q1. Do you have any of the following symptoms and how often? Please mark** | | | | | **Q2. What is the impact of these symptoms to your quality of life?** | | |
| **Symptom** | **Never** | **Rarely** | **Weekly** | **Daily** | **Multiple times per day** | **None** | **Some** | **Substantial** |
| Low mood | | | | | | | | |
| Reduced energy or activity tolerance | | | | | | | | |
| Pain on scars from chest | | | | | | | | |
| Heartburn/acid/bile (sour/bitter tasting) | | | | | | | | |
| Diarrhoea (>3 times per day) unrelated to eating | | | | | | | | |
| Bloating or cramping after eating | | | | | | | | |

### Tumour and treatment information

Information on stage, tumour location, type of surgery, postoperative complications as defined by the Esophageal Complications Consensus Group,[15] and neoadjuvant and adjuvant therapy used will be collected by reviewing medical records.

### Complications after 12 months

Specific data on long-term complications including anastomotic stricture requiring dilatation and hiatal hernia will be collected by reviewing medical records.

### Outcome data

Each symptom from the LASOR questionnaire will be graded according to impact on quality of life and frequency of the symptom, with a composite score from 0 to 5. Each EORTC HRQOL symptom item comprised four categories on a Likert scale: (1) not at all; (2) a little; (3) quite a bit; (4) very much. The previously developed LASOR symptom tool will be validated against HRQOL as measured by the EORTC QLQC30 and OG25. Patients will also be asked to complete a questionnaire (online supplementary appendix C) describing their satisfaction in completing the LASOR tool.

### Inclusion criteria

► Patients aged over 18 years at the time of surgery.
► Patients treated with oesophagectomy for oesophageal or gastro-oesophageal junctional cancer (Siewert I and II) between January 2015 and June 2019.
► Patients at least 12 months post completion of cancer treatment (surgical or oncological).

### Exclusion criteria

► Any patient who lacks capacity or is unable to provide informed consent.
► Any patient below 18 years of age at the time of surgery.
► Any patient with evidence of cancer recurrence as detected by local centres based on their own routine follow-up protocol.

### Sample size

Using simple nomograms, with a predicted prevalence of poor HRQOL of 45%,[14] based on the previously generated LASOR clinical symptom tool, to validate this tool with a sensitivity and specificity of 80%, respectively, a minimum of 640 patients would need to be recruited to the study. We anticipate an expected response rate of 80% and thus 800 patients would need to be recruited to the study.

### Statistical methodology

Each symptom from the LASOR questionnaire will graded according to impact on quality of life and frequency of the symptom, with a composite score from 0 to 5 (table 2). Each EORTC HRQOL symptom item comprises four categories on a Likert scale: (1) not at all; (2) a little; (3) quite a bit; (4) very much. Linear transformation of

**Table 2** Symptom based grading system including prevalence and impact on quality of life (QOL)—each symptom from the LAsting Symptoms after Oesophageal Resection questionnaire will graded according to impact on QOL and frequency of the symptom, with a composite score from 0 to 5

| Symptom level | QOL impact and frequency |
| --- | --- |
| 0 | No symptom present |
| 1 | QOL impact=none |
| 2 | QOL impact=some and frequency=rarely/weekly |
| 3 | QOL impact=some and frequency=daily/multiple |
| 4 | QOL impact=substantial and frequency=rarely/weekly |
| 5 | QOL impact=substantial and frequency=daily/multiple |

Likert scores for answers in each conceptual area will be performed as per EORTC recommendations. Symptom scores hence comprise a numeric value from 0 to 100, with higher scores indicating more pervasive symptoms. Poor HRQOL will be defined as having poor function and poor symptom in QLQ-C30 and QlQ-OG25 (by answering 'Quite a Bit'/'Very Much' problems to at least one question each in function and symptom scales).[16 17] Patients who do not answer 'Quite a Bit'/'Very Much' will be considered as having good HRQOL. The area under the receiver operating characteristics curve will be used as a measure of overall accuracy of the prediction model from the LASOR tool in the identification of patients with poor HRQOL as measured by EORTC QLQ-C30 and QlQ-OG25 tools.

## PATIENT AND PUBLIC INVOLVEMENT

### How was the development of the research question and outcome measures informed by patients' priorities, experience and preferences?

We presented the findings of our development study to the Oesophageal Patient Association (UK) and Heartburn Cancer UK patient support groups who identified a further three symptoms to be included in the final clinical symptom tool that we aim to validate in this present UK study.

### How did you involve patients in the design of this study?

P Behrens was a former patient and the patient representative who directly supported and contributed to the development of this protocol. His review was critical in the design of this protocol.

### Were patients involved in the recruitment to and conduct of the study?

P Behrens and patient representatives from the Oesophageal Patient Association (UK) and Heartburn Cancer

UK patient support group will be part of the steering committee for this study.

## How will the results be disseminated to study participants?
Study findings will be disseminated at Oesophageal Patient Association (UK) and Heartburn Cancer UK patient support group meetings.

## DISSEMINATION
Multiple methods of dissemination will be employed to ensure the findings from this research will reach relevant stakeholders including patients, primary care practitioners, scientists, hospital specialists in gastroenterology, oncology and surgery, health policy-makers and commissioners as well as healthcare regulatory bodies. The study findings will be presented at international gastroenterology, oncology and surgical research meetings. The findings of this research will also be presented to relevant patient groups. Ultimately, we plan to publish the results of this research in a high impact factor clinical journal to allow widespread dissemination of this research. Further as this trial will be run through the AUGIS research network, the external validity of this tool will be high within the UK population, and thus we anticipate the translation to clinical practice would be faster. Patient acceptability will also be tested and if acceptable, will further facilitate clinical implementation.

**Author affiliations**
[1]Department of Surgery & Cancer, Imperial College London, London, UK
[2]Department of Upper GI Surgery, University Hospitals Birmingham NHS Foundation Trust, Birmingham, UK
[3]Edinburgh Law School, University of Edinburgh, Edinburgh, UK
[4]Department of Surgery, Nottingham University Hospitals NHS Trust, Nottingham, UK
[5]Department of Surgery, St. Thomas' Hospital, London, UK
[6]Division of Surgery, University Hospital Southampton NHS Foundation Trust, Southampton, UK
[7]Faculty of Medicine, Department of Surgery & Cancer, Imperial College, London, UK

**Acknowledgements**  The authors also wish to thank the Oesophageal Patient Association (UK) and Heartburn Cancer UK patient support groups for the contribution to this work.

**Contributors**  SRM: study conception, design and drafting of full protocol. EAG, PS, RSV, JG, TU and GU: study design and revising full protocol. PB: patient representative involved in study design and revising full protocol.

**Funding**  This study was funded by Research Trainees Coordinating Centre (http://dx.doi.org/10.13039/501100000659) and grant number: NIHR-ACL.

**Competing interests**  None declared.

**Patient consent for publication**  Not required.

**Ethics approval**  NHS Health Research Authority (North East—York Research Ethics Committee) approval was gained 8 November 2019 (REC reference 19/NE/0352).

**Provenance and peer review**  Not commissioned; externally peer reviewed.

**ORCID iD**
Sheraz Rehan Markar http://orcid.org/0000-0001-8650-2017

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
