## [Reviewer comments · BMJ Open]

ARTICLE DETAILS

TITLE (PROVISIONAL)	Protocol for LAsing Symptoms after Oesophageal Resectional Surgery (LASORS); Multi-Centre Validation Cohort Study
AUTHORS	Markar, Sheraz; Griffiths, Ewen; Behrens, Paul; Singh, Pritam; Vohra, RS; Gossage, James; Underwood, Tim; Hanna, George

VERSION 1 – REVIEW

REVIEWER	Alexander Phillips Royal Victoria Infirmiry, Northern Oesophagogastric Unit
REVIEW RETURNED	14-Dec-2019

GENERAL COMMENTS	This protocol seeks to evaluate a new and succinct tool for looking at quality of life after oesopahgectomy. It is well written, relevant and the study likely to prove very valuable. I have no concerns with this manuscript and am happy for it to be accepted in its current form.
---

REVIEWER	BJ Noordman Erasmus MC - University Medical Centre Rotterdam
REVIEW RETURNED	Erasmus MC - University Medical Centre Rotterdam 15-Dec-2019

GENERAL COMMENTS	This is a cross-sectional study, aiming to validate a short 6-item HRQOL questionnaire to identify patients with poor HRQOL after oesophagectomy for oesophageal cancer. This tool would be clinically useful and does not yet exist. - In the introduction, the authors state that EORTC questionnaires are not designed for use in cancer-free survivorship. Nevertheless, authors use the EORTC questionnaires as reference standard to define poor HRQOL in these cancer-free patients. This is contradictory and should be explained. Did the authors consider to also use other (clinical) variables to define poor HRQOL?- It is unclear which hospitals participate, which patients will be selected, how patients will be selected and recruited, and how patients will provide informed consent. Please describe.- The authors state that 640 patients need to be recruited. It is unclear if the authors accounted for patients who do not return questionnaires. What response rate do the authors expect? How and when will non responding patients be reminded to complete questionnaires?- Additional data will collected (“archival data”). It is unclear when and why these data are collected. Please explain. Are these just (pre-treatment) base line characteristics? Do the authors aim to use these data as potential confounding factors?
---

	 - One of the objectives is to assess patient acceptability, but the methods section does not mention this. How will this be assessed? - Patients with poor HRQOL are defined as patients “answering “Quite a Bit”/“Very Much” problems to at least one question each in function and symptom scales” of the EORTC questionnaires. Is this definition based on literature? If so, please add reference(s). If not, please explain how this definition has been determined. 45% of all patients (%) are expected to have poor HRQOL using this definition. Please add references or explain. - Short-term HRQOL, pre-treatment HRQOL and HRQOL in patients with cancer recurrence will not be assessed. It should be clarified that the purpose of this study is to validate this questionnaire to identify patients with poor HRQOL one year after successful surgical treatment. - How is gastro-oesophageal junctional cancer defined? - How is evidence of cancer recurrence defined? - Is follow-up after surgery standardized? Do all participating centres/patients undergo the same follow-up protocol? Do patients routinely have (PET)-CT scans? This might have influence timing of detection of cancer recurrence (lag time bias). - Due to lag time bias, patients with undetected recurrence can be included in the study. Did the authors consider correction for recurrence within 6 months after completion of the questionnaires? - How will (severity of) comorbidities be defined? - Are complications due to (neo)adjuvant treatment, performance status, tumor histology, included as measure?
--	--

REVIEWER	Emer Guinan Trinity College Dublin, Ireland
REVIEW RETURNED	18-Dec-2019

GENERAL COMMENTS	Very timely and much needed piece of work. The methods are sound and part of a larger comprehensive project. I very much look forward to reading the results.
---

REVIEWER	Gustav Linder Department of Surgical Sciences, Uppsala University, Uppsala, Sweden
REVIEW RETURNED	27-Dec-2019

GENERAL COMMENTS	Thank you for the opportunity to review this study protocol. The authors propose a cross sectional study to validate a previously developed six symptom clinical tool (LASOR-tool) to be used in everyday practice in order to discern patients at high risk of poor health related quality of life after curative surgery for esophageal cancer. The study proposal is intriguing, very well written and the study objective, to validate a clinically useful tool, would be appealing to most clinicians working in the field of oesophageal cancer care. I have one main general concern regarding the study objective. Clinical tools have been evaluated in abundance over the years and maybe foremost tools relating to predicting risks of postoperative complications or worse outcome. In my experience these tools often become a bit incomprehensible or cumbersome and are seldom of use in the clinical setting. How do the authors propose to implement this tool if the results of the study are in
---

	favor of using the said tool? By whom and when should the assessment with the tool be made and in what way would the tool (if successful in identifying patients at risk) impact clinical decision-making? My specific comments are as follows: Methods and analysis, Archival data, page 9 row 25: How would you collect some of the demographic data suggested, such as socioeconomic status and education level, from a review of medical records? Would you be able to gather this from medical records alone or would you need other sources? Inclusion criteria, Page 10, row 13: How do you define cancer of the gastroesophageal junction? Siewert I+II but not III? Sample size, Page 10, Row 33: Please specify more clearly how the sample size calculation was performed. What test or nominal scale was used? Table 2: Explains the "composite score" outlined in the outcome data and statistical methodology section. However, it might be clearer if this is explained also in the table legend or headline so readers catch on that this is in fact the grounds for the "composite score". Nothing further.
--	---

VERSION 1 – AUTHOR RESPONSE

Reviewer: 1

Reviewer Name: Alexander Phillips

Institution and Country:

Northern Oesophagogastric Unit

Newcastle upon Tyne

Please state any competing interests or state 'None declared': None declared

Please leave your comments for the authors below

This protocol seeks to evaluate a new and succinct tool for looking at quality of life after oesophagectomy.

It is well written, relevant and the study likely to prove very valuable.

I have no concerns with this manuscript and am happy for it to be accepted in its current form.

Response: We thank you for your review of this protocol, and have made no further changes to the manuscript in response to your review.

Reviewer: 2

Reviewer Name: BJ Noordman

Institution and Country: Erasmus MC - University Medical Centre Rotterdam

Please state any competing interests or state 'None declared': None declared

Please leave your comments for the authors below

This is a cross-sectional study, aiming to validate a short 6-item HRQOL questionnaire to identify patients with poor HRQOL after oesophagectomy for oesophageal cancer. This tool would be clinically useful and does not yet exist.

- In the introduction, the authors state that EORTC questionnaires are not designed for use in cancer-free survivorship. Nevertheless, authors use the EORTC questionnaires as reference standard to define poor HRQOL in these cancer-free patients. This is contradictory and should be explained. Did the authors consider to also use other (clinical) variables to define poor HRQOL?

Response: EORTC questionnaires are the most commonly used questionnaires to evaluate HRQOL before, during and after treatment including in survivorship, despite the fact they have not been specifically designed for this purpose. We have added to introduction to clarify this [page 6 line 24].

- It is unclear which hospitals participate, which patients will be selected, how patients will be selected and recruited, and how patients will provide informed consent. Please describe.

Response: This has been added to the methods [page 8 line 6].

- The authors state that 640 patients need to be recruited. It is unclear if the authors accounted for patients who do not return questionnaires. What response rate do the authors expect? How and when will non responding patients be reminded to complete questionnaires?

Response: From our previous study [14], we had a response rate of 80%. Thus we have added this to the methods and amended the sample size calculation [page 10 line 4]. Non-responding patients will receive two reminders and if they fail to respond will be excluded from the study, we have added to the methods [page 8 line 12].

- Additional data will be collected ("archival data"). It is unclear when and why these data are collected. Please explain. Are these just (pre-treatment) base line characteristics? Do the authors aim to use these data as potential confounding factors?

Response: These data will be collected retrospectively from patient medical records, this has been added to the methods [page 8 line 17]. This data will be used to describe the cohort but not used within the analysis as this is not the aim of this study.

- One of the objectives is to assess patient acceptability, but the methods section does not mention this. How will this be assessed?

Response: We have added appendix C that details the questionnaire that will be used to assess patient acceptability, and we have added to the methods [page 9 line 12].

- Patients with poor HRQOL are defined as patients "answering "Quite a Bit"/"Very Much" problems to at least one question each in function and symptom scales" of the EORTC questionnaires. Is this definition based on literature? If so, please add reference(s). If not, please explain how this definition has been determined. 45% of all patients (%) are expected to have poor HRQOL using this definition. Please add references or explain.

Response: This is based upon EORTC guidelines for analysis, and we have added the references [page 10 line 16]. The 45% expected to have poor HRQOL is based upon data from the previous LASER study, and is now referenced [page 10 line 1].

- Short-term HRQOL, pre-treatment HRQOL and HRQOL in patients with cancer recurrence will not be assessed. It should be clarified that the purpose of this study is to validate this questionnaire to identify patients with poor HRQOL one year after successful surgical treatment.

Response: We have added this to the manuscripts global objective [page 7 line 3].

- How is gastro-oesophageal junctional cancer defined?

Response: Siewert I and II based upon upper gastrointestinal endoscopy will be included. This has been added to the methods [page 9 line 17].

- How is evidence of cancer recurrence defined?

Response: Follow-up will be carried as per the local protocol within the institution, and patients with radiological or endoscopically established recurrence will be excluded from this study. We have added to the methods [page 9 line 23].

- Is follow-up after surgery standardized? Do all participating centres/patients undergo the same follow-up protocol? Do patients routinely have (PET)-CT scans? This might have influence timing of detection of cancer recurrence (lag time bias).

Response: The detection of recurrence is not the focus of this study. As stated above, follow-up will be carried as per the local protocol within the institution, and patients with radiological or endoscopically established recurrence will be excluded from this study.

- Due to lag time bias, patients with undetected recurrence can be included in the study. Did the authors consider correction for recurrence within 6 months after completion of the questionnaires?

Response: We accept this could be a limitation of this study. However the objective of this study is to validate a tool to identify patients with poor HRQOL who may require additional support. We will ask centres to provide data at 6 months if these patients have developed recurrence, to evaluate if the questionnaire may also be able to identify those patients at risk of recurrence, but this is not the focus of the study.

- How will (severity of) comorbidities be defined?

Response: Comorbidities will be identified by their presence (yes or no) and summated to form the Charlson comorbidity index [page 8 line 20].

- Are complications due to (neo)adjuvant treatment, performance status, tumor histology, included as measure?

Response: We will include within the CRF data pertaining to ASA grade and tumour histology. However complications during neoadjuvant treatment will not be collected, as in the majority of centers neoadjuvant therapy is administered outside of the main surgical center. Data pertaining to postoperative complications will be collected and defined as per the ECCG group [page 8 line 23].

Reviewer: 3

Reviewer Name: Emer Guinan

Institution and Country: Trinity College Dublin, Ireland

Please state any competing interests or state 'None declared': None

Please leave your comments for the authors below

Very timely and much needed piece of work. The methods are sound and part of a larger comprehensive project. I very much look forward to reading the results.

Response: We thank you for your favourable review of this protocol, and have made no further changes to the manuscript in response to your review.

Reviewer: 4

Reviewer Name: Gustav Linder

Institution and Country: Department of Surgical Sciences, Uppsala University, Uppsala, Sweden
Please state any competing interests or state 'None declared': None Declared

Please leave your comments for the authors below
Thank you for the opportunity to review this study protocol.

The authors propose a cross sectional study to validate a previously developed six symptom clinical tool (LASOR-tool) to be used in everyday practice in order to discern patients at high risk of poor health related quality of life after curative surgery for esophageal cancer.

The study proposal is intriguing, very well written and the study objective, to validate a clinically useful tool, would be appealing to most clinicians working in the field of oesophageal cancer care.

I have one main general concern regarding the study objective. Clinical tools have been evaluated in abundance over the years and maybe foremost tools relating to predicting risks of postoperative complications or worse outcome. In my experience these tools often become a bit incomprehensible or cumbersome and are seldom of use in the clinical setting. How do the authors propose to implement this tool if the results of the study are in favor of using the said tool? By whom and when should the assessment with the tool be made and in what way would the tool (if successful in identifying patients at risk) impact clinical decision-making?

Response: We absolutely agree with this comment. The specific purpose of this study and the developed tool is for use in the clinic, by creating a six-symptom tool that identifies patients with poor HRQOL. Patient acceptability of this tool will be tested (Appendix B). We have added to the discussion as to how we see this tool could be implemented in clinic and replace regular postoperative assessment, and instead focus on those patients at one-year who require additional support [page 11 line 10].

My specific comments are as follows:

Methods and analysis, Archival data, page 9 row 25:

How would you collect some of the demographic data suggested, such as socioeconomic status and education level, from a review of medical records? Would you be able to gather this from medical records alone or would you need other sources?

Response: These data will be collected retrospectively from patient medical records. We have added to the methods to state this [page 8 line 17].

Inclusion criteria, Page 10, row 13:

How do you define cancer of the gastroesophageal junction? Siewert I+II but not III?

Response: Siewert I and II based upon upper gastrointestinal endoscopy will be included. This has been added to the methods [page 9 line 17].

Sample size, Page 10, Row 33:

Please specify more clearly how the sample size calculation was performed. What test or nominal scale was used?

Response: We have added to the methods to further describe the sample size calculation using simple nomograms [page 10 line 1].

Table 2:

Explains the "composite score" outlined in the outcome data and statistical methodology section. However, it might be clearer if this is explained also in the table legend or headline so readers catch on that this is in fact the grounds for the "composite score".

Response: As suggested we have added this to the Table legend [page 17 line 2].

VERSION 2 – REVIEW

REVIEWER	Bo Noordman Erasmus MC - University Medical Centre Rotterdam
REVIEW RETURNED	06-Mar-2020

GENERAL COMMENTS	All comments have been addressed adequately.
--

REVIEWER	Gustav Linder Department of Surgical Sciences, Uppsala University, Sweden
REVIEW RETURNED	06-Feb-2020

GENERAL COMMENTS	Dear Author, The revised study protocol addresses issues and concerns, raised in the initial review, to a high extent and in a satisfactory way. I have nothing further to add and therefore recommend the manuscript for publication in its present form.
---

VERSION 2 – AUTHOR RESPONSE

Reviewer(s)' Comments to Author:

Reviewer: 4

Reviewer Name: Gustav Linder

Institution and Country: Department of Surgical Sciences, Uppsala University, Sweden

Please state any competing interests or state 'None declared': None Declared

Please leave your comments for the authors below

Dear Author,

The revised study protocol addresses issues and concerns, raised in the initial review, to a high extent and in a satisfactory way. I have nothing further to add and therefore recommend the manuscript for publication in its present form.

RESPONSE: Thank you for your review.

Reviewer: 2

Reviewer Name: Bo Noordman

Institution and Country: Erasmus MC - University Medical Centre Rotterdam

Please state any competing interests or state 'None declared': Non declared

Please leave your comments for the authors below

All comments have been addressed adequately.

RESPONSE: Thank you for your review.